# ENTPD1/CD39 as a predictive marker of treatment response to gemogenovatucel-T as maintenance therapy in newly diagnosed ovarian cancer

Rodney P. Rocconi [1], Laura Stanbery[2], Min Tang[3], Luciana Madeira da Silva[4], Adam Walter[2,5], Bradley J. Monk[6], Thomas J. Herzog[7], Robert L. Coleman[8], Luisa Manning[2], Gladice Wallraven[2], Staci Horvath [2], Ernest Bognar[2], Neil Senzer[2], Scott Brun[9] & John Nemunaitis [2✉]

## Abstract

**Background** Broadened use of predictive molecular and phenotypic profiling amongst oncologists has facilitated optimal integration of targeted- and immuno-therapeutics into clinical care. However, the use of predictive immunomarkers in ovarian cancer (OC) has not consistently translated into clinical benefit. Vigil (gemogenovatucel-T) is a novel plasmid engineered autologous tumor cell immunotherapy designed to knock down the tumor suppressor cytokines, TGFβ1 and TGFβ2, augment local immune function via increased GMCSF expression and enhance presentation of clonal neoantigen epitopes.

**Methods** All patients enrolled in the VITAL trial (NCT02346747) of maintenance Vigil vs. placebo as front-line therapy with homologous recombination proficient (HRP) stage IIIB-IV newly diagnosed ovarian cancer underwent NanoString gene expression analysis. Tissue was obtained from surgically resected ovarian tumor tissue following surgical debulking. A statistical algorithm was used to analyze the NanoString gene expression data.

**Results** Using the NanoString Statistical Algorithm (NSA), we identify high expression of *ENTPD1/CD39* (which functions as the rate-limiting step in the production of the immune suppressor adenosine from ATP to ADP) as a presumptive predictor of response to Vigil versus placebo regardless of HRP status on the basis of relapse free survival (median not achieved vs 8.1 months, $p = 0.00007$) and overall survival (median not achieved vs 41.4 months, $p = 0.013$) extension.

**Conclusion** NSA should be considered for application to investigational targeted therapies in order to identify populations most likely to benefit from treatment, in preparation for efficacy conclusive trials.

## Plain Language Summary

Treatment options are limited in patients with advanced stage ovarian cancer. Treatments that stimulate the immune system to target the cancer are sometimes effective, however determining which patients will have benefit has been difficult. It is therefore important to develop new markers to predict which patients will respond to therapy. In this study, we looked at the levels of a large number of genes in tumors from patients treated with Vigil (gemogenovatucel-T), a treatment that modifies patient's own tumor cells to activate the immune system. We demonstrate that high expression of a gene named *ENTPD1/CD39* predicts a positive response to Vigil therapy. This finding could help clinicians to determine which patients might benefit from Vigil treatment and therefore might guide decisions on who should receive this treatment.

[1] University of Alabama at Birmingham, Department of Obstetrics and Gynecology, Division of Gynecologic Oncology, Mobile, USA. [2] Gradalis, Inc, Carrollton, USA. [3] Stat-Beyond Consulting, Irvine, USA. [4] Mitchell Cancer Institute, Division of Gynecologic Oncology, University of South Alabama, Mobile, USA. [5] Promedica, Toledo, USA. [6] Arizona Oncology, (US Oncology Network), University of Arizona, Creighton University, Phoenix, USA. [7] University of Cincinnati Cancer Center, Cincinnati, USA. [8] US Oncology Research, The Woodlands, USA. [9] Gold Mast Consulting, LLC, Chicago, USA. ✉email: Johnnemunaitis@gmail.com

Vigil (gemogenovatucel-T) is a novel autologous tumor cell immunotherapy, which is constructed from harvested malignant tissue[1–3]. It incorporates a multigenic plasmid encoding the human immune-stimulatory *GMCSF* gene and a bifunctional short-hairpin RNA construct, which specifically knocks down the proprotein convertase furin and its downstream targets TGFβ1 and TGFβ2[1,3,4]. It is also designed to facilitate both cancer-associated antigen and neoantigen expression, upregulate MHC-II and enhance bone-marrow derived dendritic cell maturation, thereby augmenting the afferent immune response and generating a systemic antitumor effect. The VITAL study (NCT02346747) was a Phase IIb double-blind, placebo-controlled trial involving women 18 years and older with Stage III/IV high-grade serous, endometroid or clear cell ovarian cancer (OC) in clinical complete response (CCR) following carboplatin and paclitaxel induction chemotherapy[5,6]. Results recently published in a subset of 67 patients with *BRCA1/2*-wildtype (wt) OC showed improved relapse free survival (RFS; HR = 0.51, $p = 0.02$) and overall survival (OS; HR = 0.49, $p = 0.049$) compared to placebo[5]. Moreover, ad hoc analysis of a subset of 45 patients with homologous repair proficient (HRP) tumors by Myriad MyChoice CDx (Myriad Genetics, Salt Lake City, UT) also showed improvement in RFS and OS (HR = 0.39, $p = 0.007$ and HR = 0.34, $p = 0.019$, respectively)[6]. Long term follow-up confirmed a durable survival effect[7]. Three-year survival proportion from time of procurement was 83% for Vigil and 40% for placebo ($p = 0.0006$)[7]. A correlation of systemic immune response to Vigil clinical benefit was noted using ELISPOT assay[3,8].

Contemporary clinical management of oncology patients is increasingly being guided by predictive molecular and phenotypic profiling in order to optimize the use of targeted- and immuno-therapeutics[9], e.g., tumor mutational burden (TMB), MMR, PD-1, and PD-L1[10]. However, the use of predictive biomarkers for immunotherapy in OC has not consistently translated into clinical benefit[11–16] despite documented responses in some patients[17]. Although genomically unstable, OC is not mutationally driven, thus the clinical efficacy of immunotherapy in this disease has been dismal (<10% which generally correlates with high TMB, a presumptive marker of neoantigen content), represented by several failed phase III clinical trials[11–15,18].

Nevertheless, we have studied patient subpopulations most sensitive to Vigil therapy based on molecular profile using NanoString assessment, and demonstrated that TIS^high score (tumor inflammation score) and MHC-II expression correlated with ELISPOT reactivity and clinically to OS and RFS[19]. Likewise, using NanoString technology to assess OS and RFS in patients enrolled in the VITAL study[20], we showed marked benefit in patients with *BRCA1/2*-wt and HRP profiles and improved outcomes in patients whose tumors had mutant TP53 ($p = 0.0013$). The current study explores the relationship of mRNA expression via NanoString analysis in harvested baseline tumor to RFS and OS in Vigil treated patients from the VITAL study. *ENTPD1/CD39* demonstrated clinical significance as a presumptive predictor of Vigil response versus placebo regardless of HRP status.

## Methods

**Study design and Vigil construction.** All patients provided written informed consent prior to study enrollment in the VITAL study. Briefly, the VITAL study (NCT02346747) was a phase 2b randomized, double-blind, placebo controlled trial involving women 18 years and older with stage III or IV high-grade serous, endometroid or clear cell ovarian cancer in clinical complete response. As specified in the approved clinical protocol (Mary Crowley IRB), patients provided consent for excess tissue to be used for additional immunotherapy research. Specimens were obtained

**Table 1 Demographics summary of all patients by *ENTPD1/CD39* status.**

| Characteristic | *ENTPD1/CD39* Status, No. (%) | |
| --- | --- | --- |
| | *ENTPD1/CD39* Low | *ENTPD1/CD39* High |
| No. of patients | 45 | 46 |
| Frontline chemotherapy | | |
| Neoadjuvant | 6 (13.3%) | 9 (19.6%) |
| Adjuvant | 39 (86.7%) | 37 (80.4%) |
| Stage | | |
| III | 38 (84.4%) | 39 (84.8%) |
| IV | 7 (15.6%) | 7 (15.2%) |
| Age (years) | | |
| Median (IQR) | 62.0 (56–70) | 63.5 (55–68) |
| Range | 38–79 | 42–84 |
| <65 | 27 (60%) | 26 (56.5%) |
| >= 65 | 18 (40%) | 20 (43.5%) |
| ECOG | | |
| 0 | 31 (68.9) | 30 (65.2) |
| 1 | 14 (31.1) | 16 (34.8) |
| Residual disease post-surgery | | |
| Macroscopic | 13 (28.9%) | 14 (30.4%) |
| Microscopic/NED | 32 (71.1%) | 32 (69.6%) |

from excess tissue harvested at the time of procurement for vaccine manufacture. Tissue is dissociated into cell suspension and cells are frozen at a concentration of 1.33 million cells/ml in freeze media (10% DMSO v/v in 1% HSA/plasma-Lyte A solution and stored long term in vapor phase nitrogen. Homologous recombination status [homologous recombination deficient (HRD) or HRP] was determined for all patients using the Myriad MyChoice CDx assay as previously described[6,7]. Patient demographics and CONSORT diagram are presented in Table 1 and Fig. 1, respectively.

Vigil plasmid construction and cGMP manufacturing have been previously described[5,6]. Following VITAL study protocol guidelines, ovarian tumor tissue was excised at the time of initial tumor cytoreduction surgery and shipped to Gradalis, Inc. (Dallas, TX) for tissue processing, transfection, and vaccine manufacture.

**RNA isolation and gene expression analysis.** RNA expression was determined from total RNA isolated using RNeasy Mini Kit (Qiagen, Venlo, The Netherlands). NanoString PanCancer Immuno-Oncology 360^TM CodeSet using the nCounter SPRINT platform (NanoString Technologies, Seattle, WA, USA), which includes 750 cancer expression pathway genes, was used to analyze gene expression per manufacturer protocol.

**Statistics.** For all 750 genes a NanoString statistical algorithm (NSA) was defined prior to gene analysis (Fig. 2) to assess the correlation of NanoString gene expression results with clinical benefit as measured by both RFS and OS advantage effect with Vigil to specific mRNA expression. First, a univariate Cox model was used with gene Z-scores as a continuous variable and run for both OS and RFS in Vigil treated patients. From this data, the two-sided p-value, HR and corresponding 95% confidence interval (CI) were extracted. Genes that were significant for both OS and RFS advantage at the 1% significance level were identified. The more stringent variable selection criterion of 1% significance level was used due to the relatively small number of OS/RFS events compared to the large number of genes assessed. Next, Cox proportional hazards model with interaction term for each gene identified in the univariate Cox model was used to

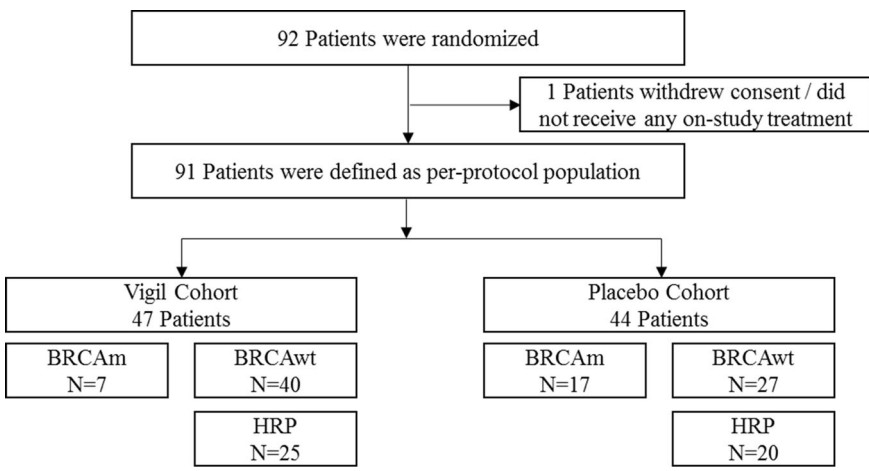

**Fig. 1 CONSORT Diagram.** Flow of patients through the VITAL trial.

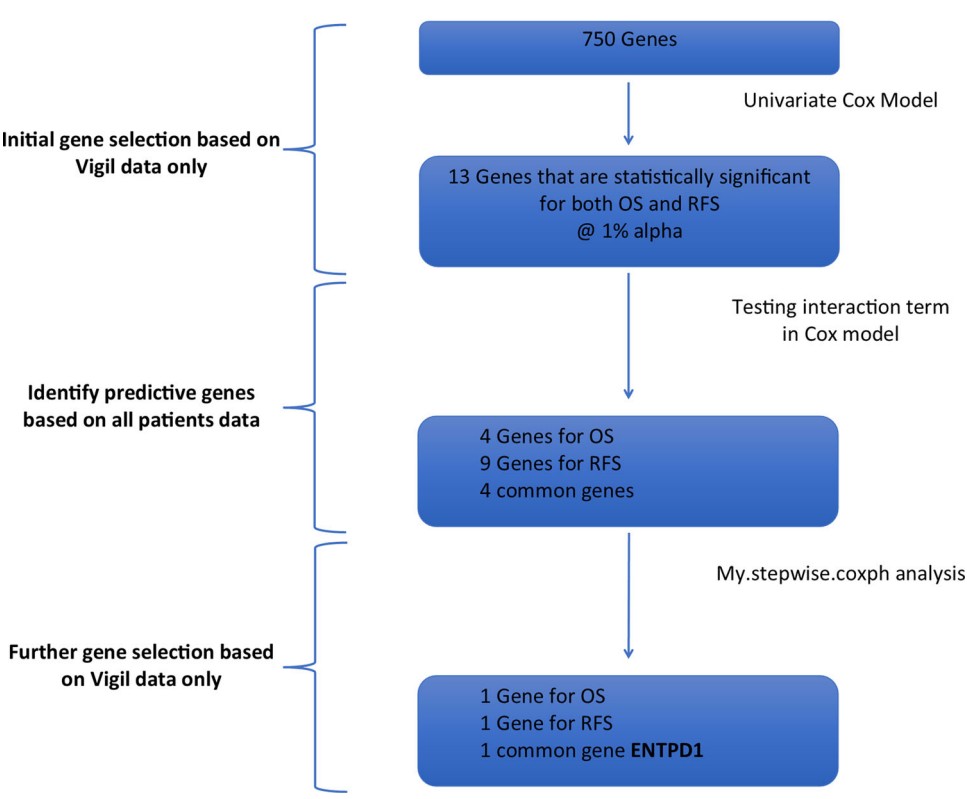

**Fig. 2 NanoString Statistical Algorithm.** Flow chart of all patients' analysis. Analyzed both with genes as raw continuous data and with genes dichotomized. Genes were selected if the interaction term was significant in both analyses. 5% alpha was used unless noted.

identify genes that were predictive of response to Vigil by analyzing data of both Vigil and placebo patients. A Cox proportional hazards model was used to determine if the interaction term between gene and treatment group was significant. The Cox model included the treatment group, gene and treatment-by-gene interaction term. The gene was considered predictive if the interaction term was significant ($p \leq 0.05$). The model was run using the gene as a continuous variable or using binary high or low gene assignment. When using binary gene assignment, the median gene value for all 91 patients was calculated for each of the 750 cancer expression pathway genes. Patients were dichotomized into high or low gene expression groups if their value was either above or below the median.

Kaplan–Meier (KM) curves were generated for genes identified as predictive for both OS and RFS. Since the identified predictive genes may not be independent, further model selection was performed using a multivariate Cox model in Vigil treated patients to further refine identification of relevant genes. We used the my.stepwise.coxph function in R (open source, R Core Team), which employs both forward selection and backward elimination methodology to further select genes that were significantly associated with the time-to-event data (OS or RFS) in Vigil treated patients[21]. The significance level for variable entry and for stay in the model was set at 0.01 and variable stay we set at 0.01 to account for potential multiplicity in the model selection process.

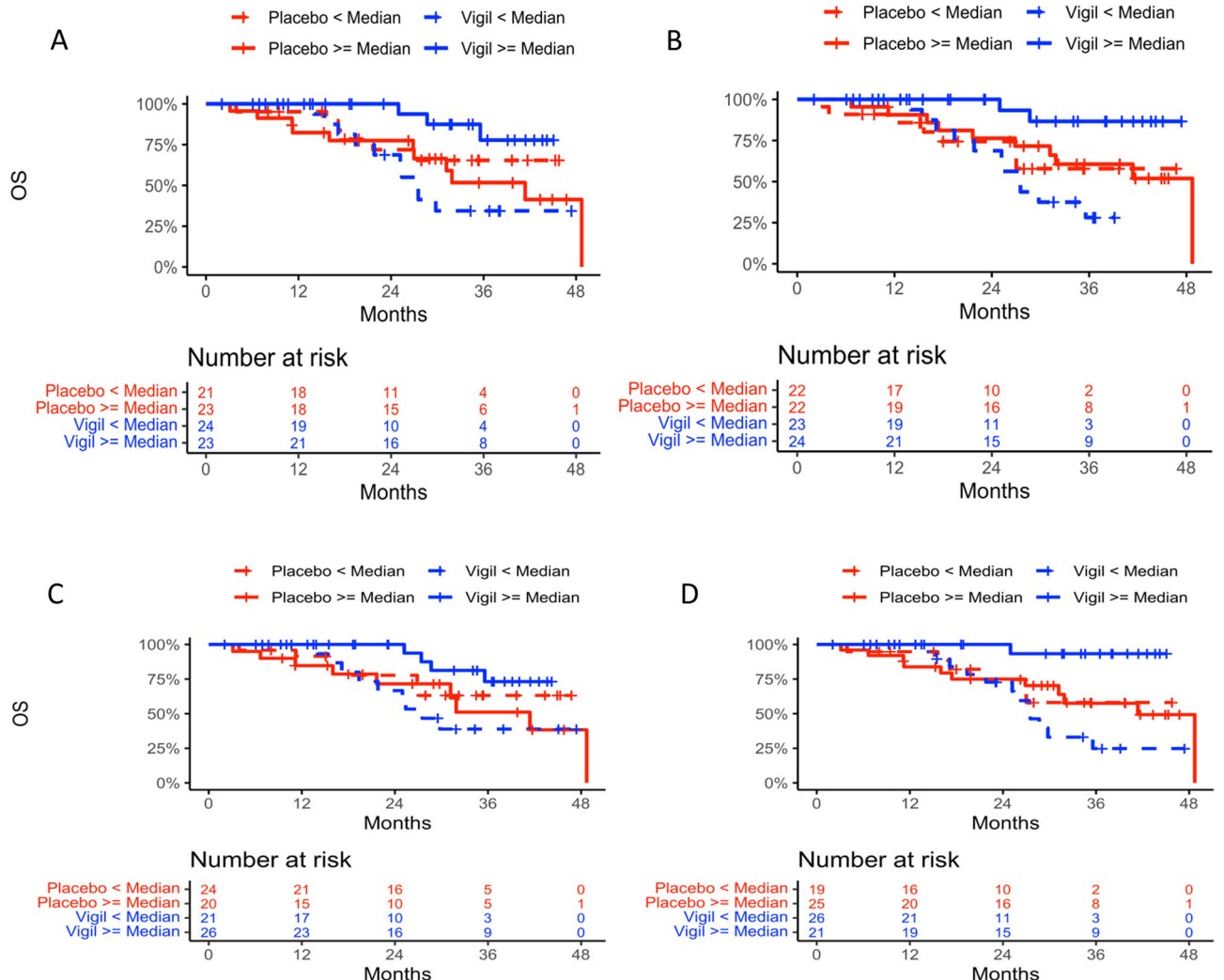

**Fig. 3 Overall survival Kaplan Meier curves of all four genes analyzed.** Overall survival Kaplan Meier curves of *ENTPD1/CD39* **A**, *CCL13* **B**, *CD79B* **C**, and *MRC1* **D** expression <median in Vigil (*n* = 24, 23, 21, 26 respectively) and placebo (*n* = 21, 22, 24, 19 respectively) and *ENTPD1/CD39*, *CCL13*, *CD79B* and *MRC1* expression ≥median in Vigil (*n* = 23, 24, 26, 21, respectively) and placebo (*n* = 23, 22, 20, 25, respectively) treated patients.

**Biomarker vigil benefit over placebo**. Previous analyses of Vigil relationship to *BRCA1/2*-wt, HRP and *TP53* mutation (*p53*-mu) subpopulations revealed correlation to clinical benefit[5–7,19,20]. These subpopulations were explored via KM analysis to assess the effect of combination biomarkers *BRCA1/2*-wt, HRP, *p53*-mu and genes identified as significant following multivariate analysis in this study on Vigil and placebo treatment effects as measured by OS and RFS.

**Reporting summary**. Further information on research design is available in the Nature Research Reporting Summary linked to this article.

## Results

**Univariate analysis vigil patients only**. First, a univariate Cox model was performed with the gene Z-score as a continuous variable to obtain the two-sided p-value, HR and 95% CI in Vigil treated patients only (n = 47). This analysis identified 13 genes that were statistically significant at the 1% significance level for both OS and RFS (Supplementary Table 1). All of these genes are associated with critical immunologic modulation function as per

NanoString Pan Cancer Immuno-Oncology 360™ Code set (NanoString Technologies, Seattle, WA, USA).

**Predictive genes using all patients data**. While the previous analysis was able to identify genes of interest, they were not able to specify if genes were predictive. To determine genes predictive of Vigil treatment efficacy, Cox proportional hazards model with interaction term was used to analyze data from both Vigil and placebo patients (n = 91). The Cox model included the treatment group, gene and treatment-by gene interaction term.

Demographics between Vigil and placebo were previously shown to not impact clinical benefit results[5,6]. Four genes were identified as predictive in both Cox models using continuous and binary data for both OS and RFS (*CD79B*, *CCL13*, *ENTPD1/CD39* and *MRC1*). Four separate KM curves were generated for each gene in: (1) Vigil patients with gene expression < median and ≥ median; (2) placebo patients with gene expression < median and ≥ median; (3) Vigil patients with gene expression < median and placebo patients < median; and (4) Vigil patients with gene expression ≥ median and placebo patients ≥ median. KM curves for OS (Fig. 3) and RFS (Fig. 4) for placebo vs. Vigil with < or ≥ median expression from these 4 genes demonstrate benefit

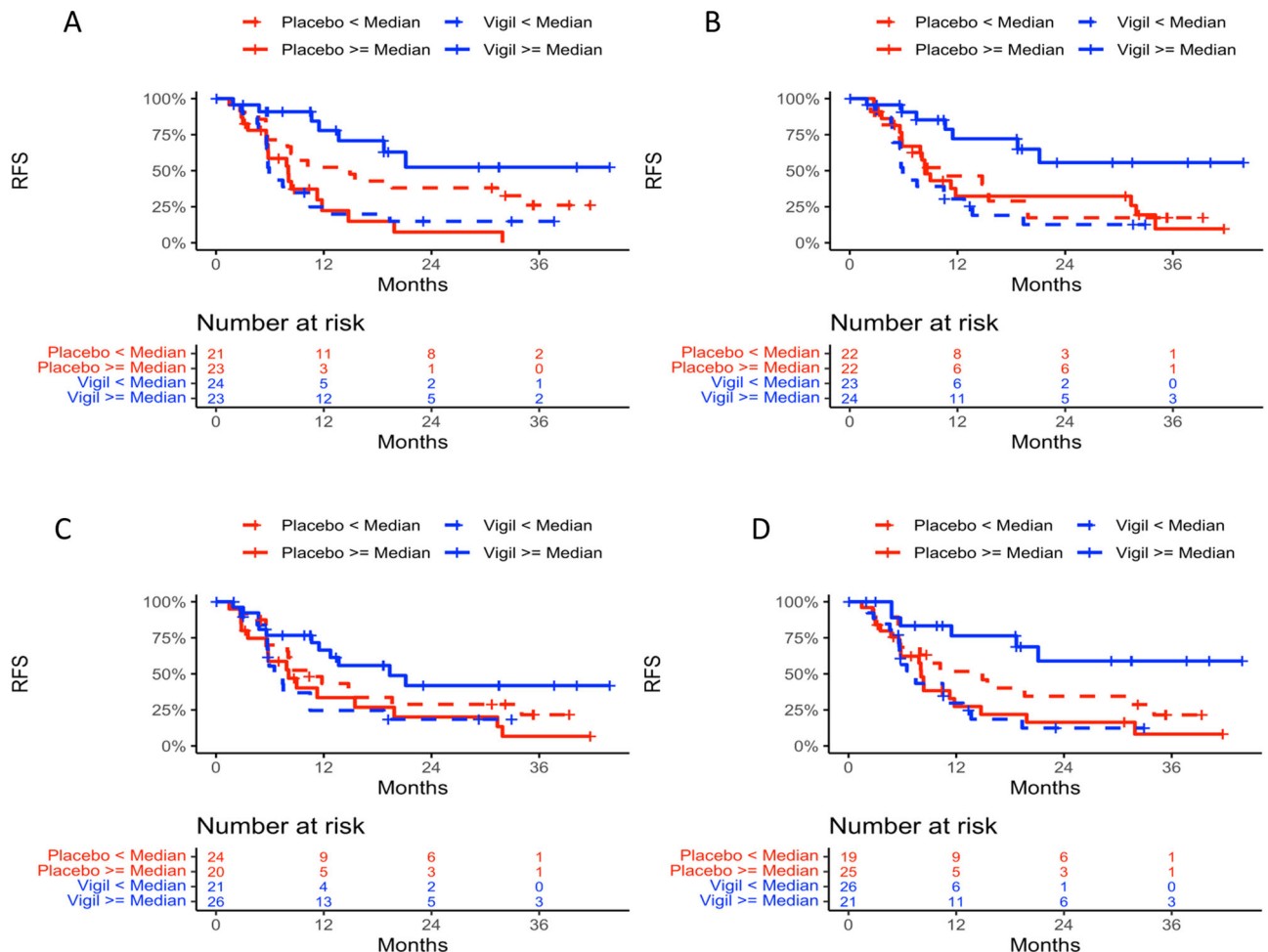

**Fig. 4 Relapse free survival Kaplan Meier curves of all four genes analyzed.** Relapse free survival Kaplan Meier curves of *ENTPD1/CD39* **A**, *CCL13* **B**, *CD79B* **C**, and *MRC1* **D** expression <median in Vigil (*n* = 24, 23, 21, 26 respectively) and placebo (*n* = 21, 22, 24, 19 respectively) and *ENTPD1/CD39*, *CCL13*, *CD79B*, and *MRC1* expression ≥median in Vigil (*n* = 23, 24, 26, 21, respectively) and placebo (*n* = 23, 22, 20, 25, respectively) treated patients. *P* values are one sided.

correlation with ≥ median expression. For patients with ≥ median *ENTPD1/CD39* expression, OS was not achieved compared to placebo OS of 41.4 months ($p = 0.013$) and median RFS was not achieved in Vigil treated compared to 8.1 months with placebo ($p = 0.00007$). Patients with ≥ median expression levels of *CCL13*, *CD79B* and *MRC1* also demonstrated OS benefit when receiving Vigil compared to placebo (median not reached vs 48.7 months, $p = 0.019$; not reached vs 41.4 months, p = 0.027; not reached vs 41.4, $p = 0.005$ respectively). Similar results demonstrating RFS benefit were observed (not achieved vs 8.4 months $p = 0.006$; 19.4 vs 8.1 months, $p = 0.010$; not achieved vs 8.1 months, $p = 0.001$ respectively). The two-sided $p$ values of the interaction term in the Cox model and one-sided p values of log rank test comparing the OS and RFS KM curves are shown in Tables 2 and 3.

**Multivariate Analysis Vigil Patients Only**. To further select significant gene associations with OS or RFS in Vigil treated patients, the my.stepwise.coxph function in R was used as the stepwise variable selection procedure (with iterations between the 'forward' and 'backward' steps) including the 4 genes showing RFS and OS advantage to Vigil treatment over placebo. Two common strategies for adding or removing variables in a multiple regression model are backward elimination and forward selection. Backward elimination begins with all genes included in the model and eliminates variables one-by-one until the model cannot be

improved per the model fitting criterion. Forward selection starts with no variables included in the model, then adds variables according to importance (e.g., based on $p$ values) until no other significant variables are found. The significance level for variable entry in the model was set at 0.01 and for variable stay was set at 0.01 to account for potential multiplicity in the model selection process. *ENTPD1/CD39* was the only gene identified through this stepwise model selection process for both OS and RFS ($p$ value < 0.001).

**Subgroup Vigil/Placebo: HRP, ENTPD1/CD39**. Twenty of the 91 patients (22%) enrolled into the VITAL trial (11 Vigil, 9 placebo) had HRP molecular profile[5–7] and *ENTPD1/CD39* "high" expression. Note HRP status and *TP53* mutations have been identified in previous analyses as predictive of Vigil response[6,7,20]. OS advantage was demonstrated (Fig. 5A) in the Vigil treated HRP/high *ENTPD1/CD39* patients relative to placebo (not achieved vs 27 months, HR = 0.23, $p = 0.025$). In the same subset, the median RFS for Vigil was 21.1 months and 5.6 months for placebo (HR = 0.18, $p = 0.004$) (Fig. 5B). Despite small sample size, these subgroup results support additional survival benefit in patients whose tumors demonstrate *ENTPD1/CD39* high expression in the HRP subgroup. In order to assess the impact of HRP and *ENTPD1/CD39* on outcomes in patients treated with Vigil, multivariate analyses including HRP status and

**Table 2 Two-sided *p* values of the interaction term in the Cox model.**

|  | Interaction term (continuous)* OS | Interaction term (continuous)* RFS | Interaction term (binary)** OS | Interaction term (binary)** RFS |
|---|---|---|---|---|
| *ENTPD1/CD39* | 0.00751 | 0.00375 | 0.0158 | 0.00014 |
| *CCL13* | 0.0190 | 0.00271 | 0.044 | 0.00998 |
| *CD79B* | 0.00426 | 0.00280 | 0.0303 | 0.0152 |
| *MRC1* | 0.01040 | 0.0169 | 0.0173 | 0.000822 |

*Analyzed with genes as raw continuous data.
**Analyzed with genes dichotomized.

**Table 3 One-sided p values of log-rank test comparing two KMs and hazard ratios and 90% CI from the univariate Cox proportional hazards model based on four predicted genes from multivariate analysis.**

|  | Vigil ≥ median vs. Vigil < median OS | | Vigil ≥ median vs. Vigil < median RFS | | Vigil ≥ median vs. placebo ≥ median OS | | Vigil ≥ median vs. placebo ≥ median RFS | |
|---|---|---|---|---|---|---|---|---|
|  | P value | HR | P value | HR | P value | HR | P value | HR |
| *ENTPD1/CD39* | 0.002 | 0.177 [0.059, 0.524] | 0.0003 | 0.238 [0.114, 0.498] | 0.013 | 0.257 [0.087, 0.761] | 0.00007 | 0.200 [0.094, 0.427] |
| *CCL13* | 0.0005 | 0.119 [0.034, 0.423] | 0.0003 | 0.236 [0.113, 0.493] | 0.019 | 0.228 [0.063, 0.824] | 0.006 | 0.338 [0.161, 0.709] |
| *CD79B* | 0.006 | 0.248 [0.092, 0.670] | 0.014 | 0.423 [0.219, 0.817] | 0.027 | 0.324 [0.118, 0.892] | 0.010 | 0.421 [0.224, 0.793] |
| *MRC1* | 0.0001 | 0.058 [0.010, 0.325] | 0.0004 | 0.229 [0.105, 0.502] | 0.005 | 0.109 [0.019, 0.613] | 0.001 | 0.245 [0.112, 0.535] |

*ENTPD1/CD39* as factors were conducted on OS and RFS for all Vigil patients. For OS, based on the multivariate Cox model for Vigil patients including both HRP status and *ENTPD1/CD39* as factors, the p values for HRP status and *ENTPD1/CD39* status are 0.30 and 0.007 respectively. For RFS, based on the multivariate Cox model for Vigil patients including both HRP status and *ENTPD1/CD39* as factors, the p values for HRP status and *ENTPD1/CD39* status are 0.15 and 0.0005 respectively. This demonstrates that within the Vigil patients, after adjusting for HRP status, *ENTPD1/CD39* is still a statistically significant factor. *ENTPD1/CD39* high Vigil patients demonstrated improved OS and RFS outcomes compared with *ENTPD1/CD39* low Vigil patients.

**Subgroup Vigil/Placebo: HRP, *p53*, *ENTPD1/CD39*.** Evidence of survival advantage was further suggested in patients with tumors demonstrating high *ENTPD1/CD39* expression and of HRP/*p53*-mu profile. Despite the small sample size ($n = 13$), a trend toward OS benefit with Vigil therapy (median not reached vs 27 months, HR = 0.34, p = 0.099) and robust RFS benefit (21.1 vs 5.6 months, HR = 0.09, p = 0.004) was suggested (Fig. 5C, D).

**Subgroup Vigil/Placebo: HRD, *ENTPD1/CD39*.** Twenty-six of the 91 patients (29%) had tumors with elevated *ENTPD1/CD39* expression that were also HRD (including *BRCA1/2*-mutation and *BRCA1/2*-wt/HRD). There appeared to be a trend towards improved OS with Vigil therapy (median not reached vs 48.7 months, HR = 0.24, p = 0.08) (Fig. 5E) and RFS difference between Vigil and placebo was highly significant in this population (median not reached vs 11.8 months, HR = 0.21, p = 0.005) (Fig. 5F).

## Discussion

Using NanoString PanCancer Immuno-Oncology 360™ molecular profiles derived from patient tumor samples in conjunction with NSA, we identified that high expression of *ENTPD1/CD39* was associated with a significant and independent improvement in OS and RFS with Vigil maintenance therapy in the VITAL study. ENTPD1/CD39 is highly expressed in OC cell-lines[22], and functions as a master regulator to maintain the balance between proinflammatory and immunosuppressive regulatory function[23].

The latter largely due the role of ENTPD1/CD39 as the rate limiting step in the conversion of ATP to ADP in the adenosine pathway. Adenosine inhibits both T-cell and NK-cell anti-tumor function. Although adenosine can be exported from the tumor into the extracellular space by nucleoside transport proteins, it is primarily formed via the action of membrane ectoenzymes by phosphohydrolysis from dead cells[24]. In addition, ENTPD1/CD39 is present on cancer extracellular vesicles (ECVs)[24]. ENTPD1/CD39 is ubiquitously expressed in the vasculature, B cells, NK cells, dendritic cells, monocytes, macrophages, regulatory T cells and monocyte derived suppressor cells in the TME[25,26]. CD8 + T cells demonstrate T cell exhaustion signatures with malignant upregulation of CD39 in the tumor microenvironment[27–29]. Moreover, T regulatory (Treg) cell upregulation of ENTPD1/CD39 within the tumor microenvironment generates immunosuppressive activity thereby facilitating malignant growth and survival[30,31]. Inhibition of ENTPD/1CD39 in murine cancer models induces anticancer activity and ENTPD1/CD39 deficient mice demonstrated a reduction in tumor growth[32–35]. Furthermore, anti-ENTPD1/CD39 increased cytotoxicity of alloreactive primed T-cell towards fresh OvCA cells[22].

In the current study, Vigil treated patients with baseline elevated tumor expression of *ENTPD1/CD39* was associated with a significantly improved response compared to those patients with tumors with low expression and to those with high tumor expression treated with placebo. The primary VITAL study results suggest that Vigil induction of *GMCSF*, knock down of TGFβ1 and TGFβ2 and induced CD8 + T cell activity targeted to tumor-specific cancer neoantigens provide anticancer activity beneficially impacts OS and RFS in newly diagnosed Stage III/IV OC patients receiving Vigil as maintenance therapy. This activity appears to be correlated to high *ENTPD1/CD39* expression—a presumptive predictive marker. Interestingly, in a murine model, high levels of TGFβ were associated with immunosuppressive CD39 + myeloid derived suppressor cells (MDSC)[36]. Notably, placebo treated patients from the VITAL study with high *ENTPD1/CD39* expression tended to show poorer survival compared to patients with lower expression, presumably reflecting the immunosuppressive role of *ENTPD1/CD39* in these patients. It is also of interest that ENTPD1/CD39 promotes tumor cell survival in hypoxic regions characterized by increased levels of ATP and

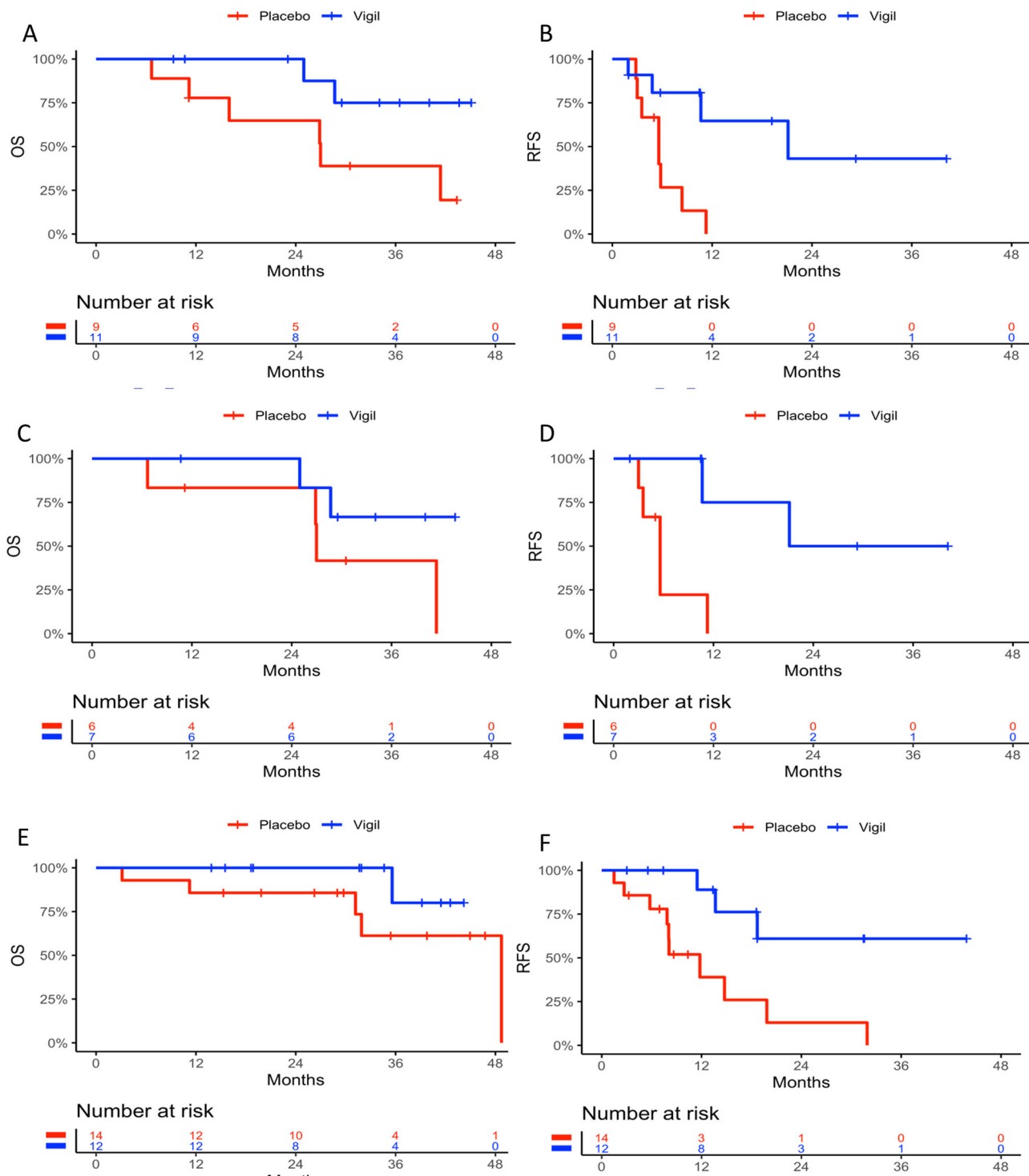

**Fig. 5 Stratification of patient population by homologous recombination and *p53* mutation status.** Kaplan Meier (KM) curves of *ENTPD1/CD39* ≥ median expression in Vigil (*n* = 11) versus placebo (*n* = 9) patients for overall survival **A** and Relapse free survival **B** in the homologous recombination proficient (HRP) population. HRP, *p53* mutant with high *ENTPD1/CD39* expression KM curves in Vigil (*n* = 7) versus placebo (*n* = 6) are presented for overall survival **C** and relapse free survival **D**. KM curves of homologous recombination deficient (HRD) patients with high *ENTPD1/CD39* expression in Vigil (*n* = 12) versus placebo (*n* = 14) for overall survival **E** and relapse free survival **F**. *P* values are one sided.

high concentrations of vascular endothelial growth factor (VEGF), thereby supporting the consideration of a combination of Vigil and a VEGF inhibitor in therapeutic trial[22].

Previously, Vigil has shown the ability to activate a systemic immune response. In Phase IIA clinical testing, all Vigil treated patients (*n* = 31) demonstrated immune activation through γIFN-ELISPOT assay which correlated with durable overall survival benefit[1,3]. Vigil also demonstrated in a small number of patients increase in the number of circulating CD3 + / CD8 + T cells following treatment[37]. In the VITAL trial we demonstrated RFS and OS benefit in patients with HRP molecular profile[6,7]. We also suggested that the presence of mutant

p53 may further improve delineation of Vigil responsive patients[20]. Results of mRNA expression via NanoString signature also indicate enhanced OS and/or RFS endpoint benefits of Vigil maintenance in both these groups. These results support the need for further verification of *ENTPD1/CD39* as a biomarker of sensitivity to Vigil treatment in OC and possibly other solid tumors with high ENTPD1/CD39 expression.

The presence of ENTPD1/CD39 in multiple cell types other than certain cancers (e.g., CD4 + / Treg, CD8 + and MDSC) supports the consideration of therapeutic assessment of combined ENTPD1/CD39 inhibition and Vigil in patients with ENTPD1/CD39[high] tumor expression. ENTPD1/CD39 monoclonal antibodies have demonstrated anticancer activity in murine models as single agents and in combination with checkpoint inhibitors and autologous EBV-specific human T cells[38]. Currently, there are a number of different CD39 targeting agents in early Phase I clinical trials under evaluation[39–41].

It is also possible that in a larger patient population receiving Vigil, supportive evidence demonstrated with the other immune modulatory signals identified by NanoString analysis (i.e., *CXCL13*[42,43], *CD79B*[44], *MRC1*[45]) will also be found to have further impact on OS and RFS. All three of these genes also perform important immunologic functions. MRC1 is expressed on tumor associated macrophages (TAMs) with M2 phenotype. Once activated MRC1 directs TAM's to M1 phenotype thereby activating the innate response[46]. Recent work has shown that high CXCL13 expression in high-grade serous ovarian cancer correlates with increased survival by maintaining CXCR5 + / CD8 + T cells with in tertiary lymphoid structures[47]. CD79b expression is limited to B cells. B cells play an important role in anti-tumor immunity through secretion of cytokines and antigen presentation[48]. Such results may further direct research towards a multiplex of biomarker sensitivity and may even direct novel combination therapeutic approaches with Vigil, including combination treatment regimens based on various molecular signal expression patterns and immune related signal pathways that are relevant to Vigil related benefit.

Molecular biomarker assessment to optimize the proportion of responsive patient populations to Vigil therapy will involve more comprehensive analyses including *p53mu*, *BRCA1/2*-wt, *ENTPD1/CD39* and HRP molecular profiles. Evidence provided in this report suggests these gene expression signals can act independently in defining sensitive subpopulations and also appears to support the possibility that the combined use of predictive biomarkers can suggest if not identify additive and possibly, synergistic therapeutic activity combinations. Clearly, statistical analyses such as those applied to the VITAL study here will likely continue to help identify optimal subpopulations with potential to benefit via treatment with Vigil as well as suggest the direction for continued Vigil combination studies. Results also justify trial consideration of Vigil in other solid tumor patients with HRP profile, *p53mu* and those with *ENTPD1/CD39* high expression by NanoString PanCancer Immuno-Oncology 360[TM] CodeSet analysis.

In conclusion, we identified gene signatures indicative of response to Vigil maintenance therapy as part of frontline treatment in newly diagnosed OC patients. Interestingly, NSA identified *ENTPD1/CD39*, a gene signal associated with an immunosuppressive tumor microenvironment, was most highly predictive of Vigil responsiveness. Previous work has indicated TGFβ may upregulate ENTPD1/CD39 in immunosuppressive myeloid cells, such that Vigil's effect in downregulating TGFβ may counter this effect and account for its activity in patients with high tumor expression of ENTPD1/CD39. Combining previously identified biomarkers of Vigil response, such as HRP and mutant *p53*, with *ENTPD1/CD39* expression may allow for refined identification of

Vigil responsive populations—ultimately allowing a transition from predictive analysis to prescriptive analytics. Such an approach can be more broadly applied to assess for correlations between gene expression signals and survival benefits as well as widen the therapeutic index by optimizing patient selection and treatment allocation with other targeted therapies.

## Data availability

Supplementary Data 1 contains source data for the main figures (Figs. 3–5) in this manuscript. Additional data that support the findings of this study, and that do not involve proprietary or similar commercial information, will be made available to appropriate parties following an approved data sharing request sent to Laura Nejedlik (lnejedlik@gradalisinc.com) at Gradalis, Inc. Requests that may involve a conflict of interest or a competitive risk maybe declined by Gradalis, Inc. in its sole discretion.

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

## Acknowledgements

We would like to acknowledge Brenda Marr for her competent and knowledgeable assistance in the preparation of the manuscript.

## Author contributions

R.P.R. was responsible for conceptualization, methodology, investigation, resources, writing – review and editing. L.S. and M.T. were responsible for conceptualization, methodology, validation, formal analysis, investigation, resources, writing – original draft, review and editing. L.M.d.S. was responsible for methodology, data generation, investigation, writing – editing. A.W. was responsible for methodology, supervision, and writing – review and editing. B.J.M., T.J.H., R.L.C., N.S. and S.B. were responsible for supervision and writing – review and editing. L.M., G.W., S.H. and E.B. contributed to investigation, resources and writing – review and editing. J.N. was responsible for conceptualization, investigation, formal analysis and writing – original draft, review and editing.

## Competing interests

The authors declare no competing interests.
