## [Peer Review File · Communications Medicine]

Reviewers' comments:

Reviewer #1 (Remarks to the Author):

The manuscript is well written and tries to look further than the classically explored biomarkers that have been used so far to guide immunotherapy. However, I have three main remarks:

1. although the trial design was published earlier, it is not clearly described in this paper and is nevertheless important for the understanding of the findings. Half of the abstract is on the clinical side of the VITAL study, then part of the intro and then again in the materials and methods. But this is always in parts and still it is not clear. So, a scheme would already clarify this and also, I feel it is better to group the information in one place. I also do not think this has to be half of the abstract, since the paper is not on the clinical outcome of the VITAL, but on the biomarkers. Also: the biological material that was studied to draft the results has to be mentioned more clearly.

2. Demographics and their link to the findings of the biomarkers is missing: Have all patients undergone primary debulking surgery? Or were there also patients with neoadjuvant chemotherapy? If yes, was there a difference (since chemo will affect the immune system)? Was there a difference between stage III and stage IV ovarian cancer patients? ...

3. I was a bit disappointed to see that there was only attention for one biomarker, nevertheless there were several more that could have been addressed (like MRC1), that would have opened better the link with the innate immune system (relevance of GM-CSF as part of the "medicinal product"). Moreover, the whole discussion and the role of the immune system is pure speculation, since nothing has been done to analyse the immune system. Nevertheless, this seems to be an important focus for the trial and is also addressed in the intro but is unfortunately not investigated (no immune monitoring of the patients). I think this is a real lack.

Reviewer #2 (Remarks to the Author):

Rocconi et al have identified a novel predictive marker of treatment to VIGIL, a clinically relevant and novel finding. A well-written paper with clearly described methods. However, some clarifications are necessary as listed below.

1) ENTPD1/CD39 is a prognostic factor as its high expression predicts better survival regardless of the treatment. Therefore the relevance of HRP/HRD status is not clear. One possibility is to use a multivariate analysis including HRP status and ENTPD1 as factors to test if HRP status provides additional information about survival in the context of ENTPD1 expression status.

2) I would also like to see the KM plot of all patients with ENTPD1 status. Sometimes subgroups can show an association in one direction, but the combined dataset can provide a different picture (Simpson's Paradox).

Reviewers' comments:

Reviewer #1 (Remarks to the Author):

The manuscript is well written and tries to look further than the classically explored biomarkers that have been used so far to guide immunotherapy. However, I have three main remarks:

1. although the trial design was published earlier, it is not clearly described in this paper and is nevertheless important for the understanding of the findings. Half of the abstract is on the clinical side of the VITAL study, then part of the intro and then again in the materials and methods. But this is always in parts and still it is not clear. So, a scheme would already clarify this and also, I feel it is better to group the information in one place. I also do not think this has to be half of the abstract, since the paper is not on the clinical outcome of the VITAL, but on the biomarkers. Also: the biological material that was studied to draft the results has to be mentioned more clearly.

We edited the abstract, introduction and methods to clarify the trial design and group the information in one place. Figure 1 (consort diagram) was added to clarify the patient populations analyzed with in the VITAL trial. The abstract was also condensed to focus on biomarkers. We also clarified that the biological material was obtained from the tumor tissue following surgical debulking.

2. Demographics and their link to the findings of the biomarkers is missing: Have all patients undergone primary debulking surgery? Or were there also patients with neoadjuvant chemotherapy? If yes, was there a difference (since chemo will affect the immune system)? Was there a difference between stage III and stage IV ovarian cancer patients? ...

We have conducted the analyses as suggested by the reviewer and added Table 1 to the manuscript. A summary is provided below.

Around 13.3% (6/45) of patients in the ENTPD1 low group had neoadjuvant chemotherapy while around 19.6% (9/46) of patients in the ENTPD1 high group had neoadjuvant chemotherapy. There is not statistically significant difference on the number of Neoadjuvant and Adjuvant patients in the ENTPD1 low and ENTPD1 high groups (p value = 0.57 per Fisher's exact test).

There are 84.4% (38/45) Stage III and 15.6% (7/45) Stage IV patients in the ENTPD1 low group. And there are 84.8% (39/46) Stage III and 15.2% (7/46) Stage IV patients in the ENTPD1 high group. There is not statistically significant difference on the number of Stage III and IV patients in the ENTPD1 low and ENTPD1 high groups (p value = 1 per Fisher's exact test).

The demographics summary for ENTPD1 high vs ENTPD1 low of all patients are provided below.

Characteristic	ENTPD1 Status, No. (%)	
	ENTPD1 Low	ENTPD1 High

No. of patients	45	46
Frontline chemotherapy		
Neoadjuvant	6 (13.3%)	9 (19.6%)
Adjuvant	39 (86.7%)	37 (80.4%)
Stage		
III	38 (84.4%)	39 (84.8%)
IV	7 (15.6%)	7 (15.2%)
Age, years		
Median (IQR)	62.0 (56-70)	63.5 (55-68)
Range	38 - 79	42 - 84
< 65	27 (60%)	26 (56.5%)
>= 65	18 (40%)	20 (43.5%)
ECOG		
0	31 (68.9)	30 (65.2)
1	14 (31.1)	16 (34.8)
Residual disease post-surgery		
Macroscopic	13 (28.9%)	14 (30.4%)
Microscopic/NED	32 (71.1%)	32 (69.6%)

3. I was a bit disappointed to see that there was only attention for one biomarker, nevertheless there were several more that could have been addressed (like MRC1), that would have opened better the link with the innate immune system (relevance of GMCSF as part of the "medicinal product"). Moreover, the whole discussion and the role of the immune system is pure speculation, since nothing has been done to analyse the immune system. Nevertheless, this seems to be an important focus for the trial and is also addressed in the intro but is unfortunately not investigated (no immune monitoring of the patients). I think this is a real lack.

The focus of this manuscript was on biomarker ENTPD1 effect of which was the only signal that fulfilled all model criteria following multivariate analysis and demonstrated significance in both RFS and OS.

Functional systemic immune response was demonstrated in relationship to Vigil and not control in Phase I and Phase IIA trial results. Results in advanced solid tumor patients including ovarian cancer patients demonstrated systemic immune activation with conversion to γ IFN ELISPOT positive in the majority of those receiving Vigil (31/31 in Phase IIA study of ovarian cancer). This correlated with immediate overall survival and long term overall survival. Additionally, we have previously demonstrated Vigil increases the amount of CD3+/CD8+ T cells. We added this to the discussion with appropriate references. Additional immune function testing and monitoring is planned with a subsequent clinical trial involving Vigil.

We also added additional discussion regarding CCL13, CD79b and MRC1 which also provided some evidence of benefit correlation but did not achieve significance once in the stepwise model. We may find that during a Phase 3 trial that testing in larger sample size that these biomarkers significance maybe gained. This will be looked at.

Reviewer #2 (Remarks to the Author):

Rocconi et al have identified a novel predictive marker of treatment to VIGIL, a clinically relevant and novel finding. A well-written paper with clearly described methods. However, some clarifications are necessary as listed below.

1) ENTPD1/CD39 is a prognostic factor as its high expression predicts better survival regardless of the treatment. Therefore the relevance of HRP/HRD status is not clear. One possibility is to use a multivariate analysis including HRP status and ENTPD1 as factors to test if HRP status provides additional information about survival in the context of ENTPD1 expression status.

We have conducted the suggested multivariate analyses and have provided a summary as below.

For OS, based on the multivariate Cox model for Vigil patients including both HRP status and ENTPD1 as factors, the p values for HRP status and ENTPD1 status are 0.30 and 0.007 respectively.

For RFS, based on the multivariate Cox model for Vigil patients including both HRP status and ENTPD1 as factors, the p values for HRP status and ENTPD1 status are 0.15 and 0.0005 respectively.

This suggests that within the Vigil patients, after adjusting for the HRP status, ENTPD1 is still a statistically significant factor with ENTPD1 high Vigil patients having better OS and RFS outcomes compared with the ENTPD1 low Vigil patients.

We have added this point to the revised manuscript.

2) I would also like to see the KM plot of all patients with ENTPD1 status. Sometimes subgroups can show an association in one direction, but the combined dataset can provide a different picture (Simpson's Paradox).

Please find below the OS and RFS KM plot of all 91 patients (45 ENTPD1 low and 46 ENTPD1 high). The combined data show a similar trend (though not statistically significant) that ENTPD1 high patients had better outcomes than the ENTPD1 low patients. For OS, the hazard ratio between ENTPD1 high and ENTPD1 low is 0.59 with 95% CI of [0.28, 1.23] and the p value of the two-sided log rank test is 0.16. For RFS, the hazard ratio between ENTPD1 high and ENTPD1 low is 0.76 with 95% CI of [0.45, 1.28] and the p value of the two-sided log rank test is 0.30.

We did not include these curves in the manuscript.

PP Population : All Patients

OS from Randomization Date

Number at risk

PP Population : All Patients

RFS from Randomization Date

Number at risk

REVIEWERS' COMMENTS:

Reviewer #1 (Remarks to the Author):

The authors have addressed all comments and the manuscript has been adapted in the appropriate way. I think the manuscript has improved and therefore, I have no further comments.

Reviewer #2 (Remarks to the Author):

None